# MEMOIZATION-AWARE BAYESIAN OPTIMIZATION FOR AI PIPELINES WITH UNKNOWN COSTS

## ABSTRACT

Bayesian optimization (BO) is an effective approach for optimizing expensive black-box functions via potentially noisy function evaluations. However, few BO techniques address the cost-aware setting, in which different samples impose different costs on the optimizer, particularly when costs are initially unknown. This cost-aware BO setting is of special interest in tuning multi-stage AI pipelines, in which we could apply caching techniques to store and reuse early-stage outputs in favor of optimizing later stages, without incurring the costs of re-running the full pipeline. In this paper, we propose the Expected-Expected Improvement Per Unit Cost (EEIPU), a novel extension to the Expected Improvement (EI) acquisition function that adapts to unknown costs in multi-stage pipelines. EEIPU fits individual Gaussian Process (GP) models for each stage's cost data and manages the different cost regions of the search space, while balancing exploration-exploitation trade-offs. Additionally, EEIPU incorporates early-stage memoization, reducing redundant computations and costs by reusing the results of earlier stages, allowing for more iterations than existing approaches within the specified budget. In the cost-aware setting, EEIPU significantly outperforms comparable methods when tested on both synthetic and real pipelines, returning higher objective function values at lower total execution costs. This offers a significant advancement in cost-aware BO for optimizing multi-stage machine learning pipelines.

## 1 INTRODUCTION

Machine Learning (ML) is a complex practice that extends beyond the final tuning of a trainable model, but is rather a multi-stage process stretching from data collection to deployment. Due to the cascade nature of ML or AI pipelines (*i.e.* the sequence of steps in the preparation, training, or deployment of a model) where each step depends on previous stage outputs, optimization must be performed on every stage to achieve the desired model performance. However, many stages in a typical AI pipeline have no known analytical form, and thus cannot be optimized using non-black-box techniques. This motivates Bayesian Optimization (BO), a black-box optimization method popular for tasks whose objective functions are expensive and do not admit gradient information. BO relies on potentially noisy function evaluations $f(x_1), f(x_2), ..., f(x_N)$ to create a surrogate model of the function, thus facilitating sample-efficient optimization.

In BO, surrogate function modeling is typically achieved through a Gaussian Process (GP) model (Rasmussen, 2004), which relies on an acquisition function for choosing the next evaluation point. As one example, the Expected Improvement (EI) acquisition function Mockus et al. (1978) has become a popular choice for minimizing the number of function evaluations by effectively balancing exploration and exploitation (Frazier, 2018). EI is defined as the expected improvement a potential observation has over the current optimal objective value, and is defined as $\text{EI}(x) = \mathbb{E}[\max(f(x) - f(x^*), 0)]$.

Many standard BO techniques rely on the number of evaluations as a sole efficiency metric. While optimizing for this metric is effective at reducing the number of optimization iterations, it works under the assumption that evaluation costs are uniformly distributed across the function domain, which does not hold in practice (e.g. depth of a decision tree could exponentially increase time complexity in the worst case). It is, however, challenging to account for other costs during evaluation, since it is usually impossible to know the cost of an evaluation before it is run for the first time. Previous works have considered this issue, showing that EI behaves suboptimally when its performance is measured by cost, rather than number, of evaluations. Some promising approaches, such as the

Expected Improvement Per Second (EIPS) (Snoek et al., 2012) and Cost Apportioned BO (CArBO) (Lee et al., 2020), tackle this issue by proposing to fit a second GP to model log-costs $\ln c(x)$ under the assumption of positive costs $c(x) : \mathcal{X} \to \mathbb{R}^+$, alongside $f(x)$. These approaches, however, lack details on the implementation of their proposal on real models.

Cost awareness becomes more important when tuning real-world AI pipelines, which frequently contain multiple stages. Different pipeline stages incur different costs, and inter-stage dependency—where each stage depends on previous outputs before being processed—creates, at first glance, a bottleneck where the entire pipeline needs to run to completion before the acquisition function proposes a new candidate point. Previous methods have generalized Bayesian Optimization to a multi-stage setting (Kusakawa et al., 2021), attempting to optimize runtime by proposing a mechanism to suspend pipelines in the middle of a multi-stage decision-making process; however, these methods cannot function when costs are unknown.

In this paper, we propose the Expected-Expected Improvement Per Unit-cost (EEIPU) acquisition function, illustrated in Figure 1, as an extension of EI that combines cost-aware Bayesian Optimization with memoization—i.e. the ability to partially save and resume pipelines, thus lowering the cost of rerunning the full pipeline—for the multi-stage setting. This allows for significantly more iterations than comparable methods within the specified cost budget. EEIPU works by fitting a GP to model the objective function $f$, as well as a separate log-cost GP for every stage of the pipeline. Per our knowledge, there has been no published work that adapts cost awareness as well as memoization awareness to multi-stage pipeline optimization with unknown costs. By applying memoization to the pipeline, EEIPU reduces the cost of subspaces of evaluation points $x \in \mathcal{X}$ that match an already-executed prefix of the pipeline stages, creating new opportunities for cost-efficient exploration. Our main contributions can be summarized as follows:

- We propose EEIPU, a cost-aware acquisition function for pipeline settings, which extends Expected Improvement (EI). After computing $\text{EI}(x)$ for candidate points, EEIPU uses the log-cost GP models to compute the expected total inverse cost, where the total cost $C(x)$ is defined as the sum of individual stage costs $c_i(x)$ of the pipeline. EEIPU is then defined as $\text{EEIPU}(x) \triangleq \mathbb{E}[\text{EI}(x)/C(x)] \triangleq \text{EI}(x) \times \mathbb{E}[1/C(x)]$.

- Our proposal combines cost awareness with sample efficiency by assigning a cost cooling factor $\eta$ to the expected inverse cost, which sequentially decreases as a function of the provided cost budget. This budget-based cost-cooling approach, proposed by Lee et al. (2020), aims to set a cost-effective initial design that exploits low-cost regions, before sequentially encouraging the algorithm to explore higher cost regions for potentially optimal candidate points when the low-cost search space is sufficiently explored.

- EEIPU is memoization-aware: it caches the output of each pipeline stage. By reusing earlier stage outputs, EEIPU reduces the cost of exploring parameter regions in later stages.

- We run EEIPU on a wide range of experiments on both synthetic and real multi-stage pipelines. Our results find that EEIPU significantly outperforms existing BO baselines by achieving similar, if not higher, objective values, while only incurring a fraction of the runtime costs.

## 2 BACKGROUND

An early work addressing cost-aware black box function optimization is Snoek et al. (2012), who introduce the expected improvement per second (EIPS) acquisition function. Their innovation is the incorporation of a duration function into the traditional EI acquisition function. The logarithm of said duration function is used as a a cost function, and modeled with a Gaussian process. The scaling of EI by the cost function term allows the acquisition function to account for the differences in wall-clock time that be incurred by different evaluated points in the hyperparameter search space.

Lee et al. (2020), however, in their work on Cost-Aware Bayesian Optimization (CArBO), rename EIPS as EIPU, making the observation that this method often underperforms relative to EI. They reimagine $\text{EIPU}(x) = \text{EI}(x)/c(x)^\eta$ by introducing an exponent $\eta$ term into the cost function in the denominator. This eta factor, as it decays, reduces EIPU to EI, in effect allowing for expensive evaluations to be explored after cheaper ones have been fully exploited. While the results were promising, both techniques proposed by Snoek et al. (2012) and Lee et al. (2020) handle BO in the unknown-cost setting by fitting a cost GP to a single pipeline stage, which prevents them from

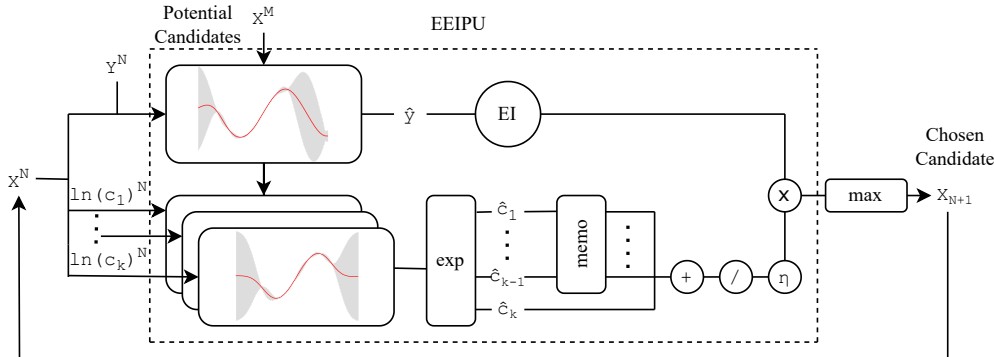

Figure 1: EEIPU Pipeline Architecture. Given a set of N observations, their corresponding objective values, the costs of running them on each stage of the $k$-stage pipeline, and the remaining optimization budget, we fit $k + 1$ GP models. During each BO iteration, EEIPU generates $M$ potential candidates $X^M$, and estimates their objective and per-stage costs by sampling from the GP models. Then, (1) $\forall x \in X^M$, the objective GP samples are used to estimate $\text{EI}(x)$, and (2) the samples from the $k$ cost GPs are used to estimate the expected costs. Memoized stages, if any, have their costs discarded by the *memo* gate, leaving only non-memoized stages to compute expected inverse cost $\mathbb{E}[1/C(x)]$, which is raised to the power $\eta$ for a cost-cooling mechanism (Lee et al., 2020). The product of (1) and (2) defines the EEIPU acquisition function, which is maximized to return the chosen candidate for the current BO iteration. See Section 3 for details.

taking advantage of memoization in multi-stage pipelines. Abdolshah et al. (2019) also address a single-stage setting, but in a specific context where search space dimensions are rank-ordered by a known cost. Here, they address cost awareness in a multi-objective (CA-MOBO) setting with a non-uniform cost landscape by specifying cost-aware constraints over the search space. The cost constraints are specified as a sorted tuple of indexes that indicate the order of dimensions of the search space according to the cost of changing the values at that dimension. CA-MOBO, thus, constrains the acquisition function to search inexpensive regions by fixing lower values for the cheaper regions while searching the space of the more expensive region. However, this approach requires foreknowledge of the cost landscape and the relative change in costs with respect to the change in the dimensions.

Cost evaluation in BO with an unknown cost landscape has also been studied by some recent works. Astudillo et al. (2021) develop a strategy that factors cost as an unknown variable in a cost budget-constrained setting. The paper not only considers the cost of evaluating an objective function but also the cost of identifying the next point to evaluate as part of the total budget using a non-myopic policy. Likewise, Zhang et al. (2022) also attempts to solve the hyperparameter optimization problem with a cost-budget, using a non-myopic searching technique that considers both the past knowledge of the objective and cost functions and the effect of depleting the budget based on the balance available.

Two important works (Lin et al., 2021; Kusakawa et al., 2021) have attempted to address the multi-stage setting. The LaMBO technique of Lin et al. (2021) models the objective function as a GP and optimizes the acquisition function with cost awareness to query the next point for evaluation. The paper models the multi-stage cost awareness as a multi-armed bandit problem, whereby switching costs between two different arms (or hyperparameter variables) incurs a cost. This technique simulates a memoization-like process when switching between bandit arms, through making variables in earlier stages of the pipeline more costly to evaluate, hence opting for a cheaper path of only evaluating later stages. However, this approach assumes that the cost function is known in all partitions of the search space. This leaves open the problem of how to deal with non-theoretical scenarios in which a cost or duration function is unknown, and how that changes when applied in a multistage process.

Kusakawa et al. (2021) refer to the multi-stage setting as a cascade process. They develop an efficient technique for suspending the multi-stage decision-making process in the middle of the cascade when poor-performing candidates are detected. While their work aims to address the problem of high evaluation costs at each stage, they also assume that costs are known. Furthermore, they assume that the cost for each stage is fixed. In our setting, the cost at a given stage is a modeled as a function of the hyperparameters corresponding to that stage.

## 3 METHODOLOGY

In this section, we detail our EEIPU method. This approach can broadly be categorized into three interconnected components: cost awareness, memoization awareness, and cost cooling. We provide a detailed description of each mechanism, which are combined to define the EEIPU framework as shown in Figure 1. Because our method follows an iterative process, we will explain the components through a single BO iteration for choosing an optimal set of parameters $\mathcal{X}_i$ to run through a $k$-stage pipeline, where $s_j$ is used to refer to each stage, $x_{ij}$ is the set of parameters in $\mathcal{X}_i$ corresponding to $s_j$, $y_{ij}$ is the output of running $\mathcal{X}_i$ through $s_j$, which is then fed to $s_{j+1}$, until the final output $\mathcal{Y}_i$ is obtained. $c_{ij}$ refers to the cost of running $\mathcal{X}_i$ through stage $s_j$ of the pipeline.

### 3.1 COST AWARENESS WITH UNKNOWN COSTS

Snoek et al. (2012) extend EI to incorporate cost awareness in a single-stage setting as

$$\text{EIPS}(x) := \frac{\text{EI}(x)}{c(x)} \tag{1}$$

which was then improved by Lee et al. (2020), who redefined cost awareness in a more robust way as

$$\text{CArBO}(x) := \frac{\text{EI}(x)}{c(x)^\eta} \tag{2}$$

where $c(x)$ is the cost of evaluating the observation $x$, and $\eta$ is a cost-cooling factor. Snoek et al. (2012) and Lee et al. (2020) briefly mention the cost modeling process for handling unknown costs, but provide no details about the inverse cost estimation nor a source code to reproduce their results. Therefore, for our experiment setting, we adapt EIPS and CArBO to our unknown cost setting to evaluate the advantage of combining cost with memoization-awareness.

We also leverage our synthetic experiments to define EIPU-MEMO, an oracle version of the memoization-aware EEIPU that is granted knowledge of the true cost function of the pipeline. Note that this definition of EIPU-MEMO does not meet the requirements of our cost-aware setting, in which the cost function is unknown and must be learned. EIPU-MEMO serves as a theoretical version of EEIPU to verify the the intended behavior of our approach beyond the randomness that comes with modeling costs using GP models. Details on this experiment can be found in Appendix D.

To make our method practical in a real setting, we further extend the aforementioned EIPU-MEMO to a multi-stage setting, and provide a detailed approach to estimating the expected total inverse cost. With unknown evaluation costs, we leverage the GP setting that we fit on $\{\mathcal{X}, \mathcal{Y}\}$ and apply it to costs alongside objectives. When a new observation $\mathcal{X}_i$ is chosen by the acquisition function, it is run through the multi-stage pipeline to obtain the final objective value $\mathcal{Y}_i$, which is added to our set of observations for model retraining in the following iteration. Additionally, we keep track of the costs of running $\mathcal{X}_i$ through each stage of the pipeline, which serves to build $k$ cost datasets, which are then used to fit $k$ separate cost-GP models on $\{\mathcal{X}, \ln(c_j)\}, \forall j \in \{1, 2, ..., k\}$ for cost prediction. Under the assumption of positive costs, GP models are fit to log-costs rather than raw values. Positivity is then enforced through predicting costs as $c_j = \exp(\ln(c_j))$. For the purposes of this paper, we assume that the costs of different stages of the pipeline are independent of each other when modeling the stage-wise cost-GPs.

We thus define the *Expected-Expected Improvement Per Unit-cost (EEIPU)* acquisition function to be $\text{EEIPU}(x) \triangleq \mathbb{E}[\text{EI}(x)/C(x)]$, i.e., the expected EIPU acquisition, or equivalently the product of expected improvement (EI) and the expected total inverse cost $\mathbb{I}(x) = \mathbb{E}[\frac{1}{C(x)}]$. Initially, a batch of $M$ potential candidates $\{X\}$ is generated using BoTorch's `optimize_acqf()` function (Balandat et al., 2020) and passed to EEIPU. The process of returning the candidate $\mathcal{X}_m$ with the highest EEIPU value begins with a standard computation of $\text{EI}(\mathcal{X}_m)$ (Mockus et al., 1978). Then, because $\mathbb{E}[\frac{1}{C(x)}] \neq \frac{1}{\mathbb{E}[C(x)]}$, and because our cost-GP models are fit on individual stages in order to allow for independent stage-cost discounting during the memoization process, the inverse cost is therefore estimated by running the following process on each candidate $\mathcal{X}_m$:

1. We apply Monte Carlo sampling to retrieve a batch of $D$ predictions from every log-cost GP, where $D$ is chosen to represent a fair and unbiased estimation of the stage cost.

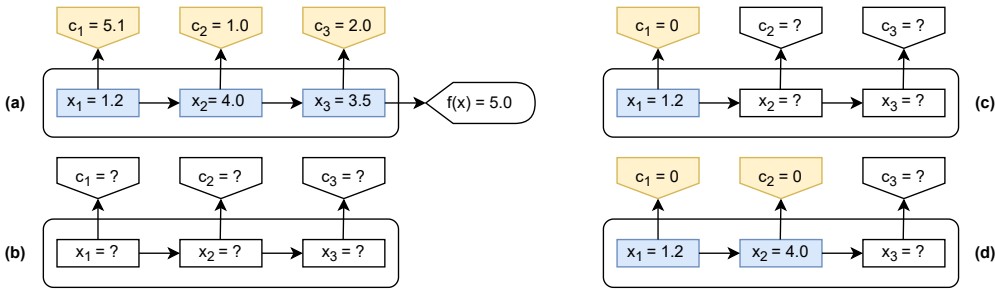

Figure 2: Prefix pooling for 1 observation (a) in a 3-stage pipeline. Each stage has a single parameter, respectively represented by $x_1, x_2$, and $x_3$, with corresponding costs $c_1, c_2, c_3$. Since the query in (a) is completed, both the cost and output value of each stage are stored. The first 2 stages of the observation are cached as prefixes (c) & (d) in order to avoid rerunning them if need be. "?" indicates un-memoized stages, and the empty prefix (b) is used for open-ended search. This process is independently applied to every observation chosen by the acquisition function.

2. After sampling $D \times k$ predictions for each candidate $\mathcal{X}_m$, we compute the total cost for each sample as $C_d = \sum_{j=1}^{k} c_{d,j}$, for all $d \in \{1, 2, ..., D\}$.

3. The total inverse cost is then estimated as:

$$\mathbb{I}(x) = \mathop{\mathbb{E}}_{C(x)\sim\mathcal{N}(\mu, K(x, x'))} \left[ \frac{1}{C(x)} \right] \approx \frac{1}{D} \sum_{d=1}^{D} \frac{1}{C_d}. \tag{3}$$

## 3.2 MEMOIZATION AWARENESS

In this section, we discuss our memoization approach, where we cache stage-wise outputs of previously evaluated candidate points, in order to bypass the need of rerunning prefix stages when attempting to further optimize later stages of the pipeline.

Given the task of hyperparameter tuning, we start with $N$ previously evaluated hyperparameter combinations. Each hyperparameter combination is divided into $k$ subsets, every one of which corresponds to a separate stage of the pipeline. A set of $N \times (k - 1)$ prefixes are cached, such that the corresponding stage outputs can easily be retrieved by using the stage's hyperparameters as a querying key. The caching process of a 3-stage pipeline is explained in Figure 5, and works by memoizing all $k - 1$ prefix stages of an observation $\mathcal{X}_i$, excluding the complete set. In other words, we memoize an observation $\mathcal{X}_i$ by caching the following configurations: $\{[(x_{i,1}, y_{i,1})], [(x_{i,1}, y_{i,1}), (x_{i,2}, y_{i,2})], ..., [(x_{i,1}, y_{i,1}), (x_{i,2}, y_{i,2}), ..., (x_{i,k-1}, y_{i,k-1})]\}$, This separate database is built and stored for every evaluated candidate point, in addition to the empty prefix $[()]$, which is used to encourage exploring entirely new configurations. Let this database be referred to as $P_X$

When generating $M$ candidate points for EEIPU computation, as detailed in Section 3.1, we equally divide $M$ into a random batch of subsets of $P_X$, in addition to the empty subset $\{[()]\}$. Prefix values of each candidate point are then fixed to the values of their corresponding cached stages, before randomly assigning values to the later, unmemoized stages. This process ensures that the empty prefix serves to explore entirely new areas of the search space, while the other points exploit previously seen areas, for a chance of a potentially higher objective value at a fraction of the cost.

The $M$ potential candidates are then run through EEIPU to compute the expected-expected improvement per unit cost of each of the generated configurations, as per Section 3.1. To incorporate memoization into the process, we pass a $\delta$ parameter to the forward function for cost discounting, which stands for the number of cached stages for each configuration being evaluated. When running a candidate $\mathcal{X}_m$ through EEIPU, we sample expected costs from GP models $\delta + 1$ through $k$, whereas the expected costs for stages 1 through $\delta$ are set as an $\epsilon$-cost to account for overhead while discounting the cost of running the full stages of the pipeline, and leveraging their cached outputs instead without the need of sampling their expected costs. More formally, the total cost of a potential candidate $\mathcal{X}_m$ is set by EEIPU to be:

$$\mathbb{E}\left[C(x)\right] = \mathbb{E}\left[\sum_{j=1}^{k} c_j(x)\right] = \sum_{j=1}^{k} \mathbb{E}\left[c_j(x)\right] \approx \sum_{j=1}^{\delta} \epsilon + \sum_{j=\delta+1}^{k} \mathbb{E}\left[c_j(x)\right], \tag{4}$$

where the stagewise costs $c_j(x) \sim \mathcal{N}(\mu, K(x, x'))$. This cost discounting process serves to assign a high level of importance to cached configurations, such that, when compared to another un-memoized configuration with a relatively close expected improvement, priority would be assigned to cheaper evaluations, especially during earlier iterations of the optimization process.

Following the prefix cost discounting in Eq. (4), applied to the expectation in Eq. (3), EEIPU incorporates cost awareness by defining $\mathbb{I}(x)$ as the expected inverse cost after discounting, and the expected-expected improvement per unit cost of a candidate $\mathcal{X}_m$ as:

$$\text{EEIPU}(x) = \text{EI}(x) * \mathbb{I}(x) \tag{5}$$

### 3.3 Cost Cooling

The main aim of our method is to incorporate cost awareness to redefine expected improvement in a practical setting. However, the standard EI has an undeniable advantage in choosing candidates with the highest expected improvement over objective value, and prioritizing low-cost regions throughout the optimization process may eventually lead to compromising on objective value in favor of low evaluation costs.

To tackle this issue, we adopt a budget-based cost-cooling mechanism to assign a high level of importance to low-cost regions during earlier iterations of the process, before gradually pushing EEIPU towards exploring higher-cost regions in later iterations of the process, as the assigned budget decreases. We implement this by using an annealing schedule on the cost, inspired by the *EI-COOL* method (Lee et al., 2020). Cost cooling is defined by a factor $\eta$, which applies an exponential decay to the expected inverse cost $\mathbb{I}(x)$. This factor is set to an initial value $\eta_0 = 1$, which is then decayed every iteration with respect to the remaining optimization budget, such that: $\eta_i = \frac{\text{remaining budget}}{\text{total budget}}$. This adds the following update to the EEIPU computation in Eq. (5):

$$\text{EEIPU}(x) = \text{EI}(x) * \mathbb{I}(x)^\eta \tag{6}$$

### 3.4 EEIPU Framework and Main Contributions

The components detailed above make up the EEIPU method, illustrated in Figure 1. EEIPU is the first multi-stage BO approach to incorporate cost and memoization awareness with unknown costs. Table 1 highlights our main contribution over existing methods.

When comparing EEIPU against the most related methods listed in Table 1, we found that none of those methods, other than EI (Mockus et al., 1978), provided source code for their proposals. Therefore, we chose the most adaptable approaches to EEIPU, which are CArBO (Lee et al., 2020) and EIPS (Snoek et al., 2012), and adapted them to our unknown-cost setting. While LaMBO (Lin et al., 2021) implements a memoization variant, it deals with a fairly different setting of switching costs than our proposed method approaches the problem. Because the paper did not provide sufficient information to reproduce the algorithm and problem setting, and the paper authors did not respond to our request for code, implementing LaMBO would therefore require a complete switch of our problem setting and how we set our environment. In our experiments, we aim to identify the potential of cost awareness and memoization awareness on synthetic as well as real AI pipelines by running EEIPU against the EI, CArBO, and EIPS acquisition functions. The pseudocode of our EEIPU algorithm detailing the prefix sampling and the EEIPU computation processes is defined in Algorithm 1.

Table 1: Main EEIPU Contributions Relative to Comparable BO Methods

| Acquisition Function | Cost-Aware | Unknown Costs | Multi-Stage | Memoization-Aware |
|---|---|---|---|---|
| EI (Mockus et al., 1978) | No | No | No | No |
| CArBO (Lee et al., 2020) | **Yes** | **Yes** | No | No |
| EIPS (Snoek et al., 2012) | **Yes** | **Yes** | No | No |
| Multi-stage BO (Kusakawa et al., 2021) | No | No | **Yes** | **Yes** |
| LaMBO (Lin et al., 2021) | **Yes** | No | **Yes** | **Yes** |
| EEIPU | **Yes** | **Yes** | **Yes** | **Yes** |

## 4 Experiments

In this section, we define our synthetic experiment setting, where we tested **EEIPU** against **CArBO, EIPS,** and **EI** on a set of benchmark functions, typically used for standard BO practices in previous

---

**Algorithm 1** Maximizing EEIPU (Expected-Expected Improvement Per Unit) Acquisition Function

---

**Require:** All prefix stages in the evaluated dataset $P$, $\text{GP}_y$, $\text{GP}_{ci} \forall i \in \{1, 2, ..., K\}$, $M$, $D$, $\texttt{lo\_bounds}$, $\texttt{hi\_bounds}$, $\texttt{remaining\_budget}$, $\texttt{total\_budget}$
  $N = \texttt{len}(P), \texttt{n\_cand} = M/N$
  **for** (n_stages, pref) in $P$ **do**                  ▷ Iterate over all memoized hyperparameter prefixes
     $\delta \leftarrow \texttt{len}(\text{pref}), \texttt{lo\_bounds}[: \delta] \leftarrow \text{pref}, \quad \texttt{hi\_bounds}[: \delta] \leftarrow \text{pref}$
                            ▷ Monte Carlo sampler draws samples that prefix-match the prefix values
     $\text{cands} = \texttt{generate\_random\_candidates}(\delta, \texttt{n\_cands}, (\texttt{lo\_bounds}, \texttt{hi\_bounds}))$
     **for** $i$ in range($\texttt{len}(\text{cands})$) **do**              ▷ Maximize EEIPU via Monte Carlo sampling
        $X \leftarrow \text{cands}[i], \hat{Y} \leftarrow \text{GP}_y(X, D)$      ▷ $D$ is the number of sampled predictions per GP sampling
        **for** $s$ in n_stages **do**    $\hat{C}_s \leftarrow \epsilon$ **if** $s < \delta$ **else**    $\hat{C}_s \leftarrow \text{GP}_{cs}(X[: s], D)$
                                 ▷ Discount the cost of memoized stages
        **end for**
        $\text{EI} \leftarrow \texttt{compute\_expected\_improvement}(\hat{Y})$
        $\mathbb{I} \leftarrow \texttt{compute\_expected\_inverse\_costs}(\hat{C})$
        $\eta \leftarrow \frac{\texttt{remaining\_budget}}{\texttt{total\_budget}}$
        $\text{EEIPU}[i] \leftarrow \text{EI} \times \mathbb{I}^{\eta}$
     **end for**
     return $\texttt{argmax}(\text{EEIPU})$
  **end for**

---

works (Kirschner et al., 2019; Vellanki et al., 2017). Then, we ran the same set of acquisition functions on a stacking, as well as a segmentation pipeline to demonstrate and evaluate the practical uses of EEIPU in real AI pipelines. To ensure unbiased results, we ran every experiment on $T = 10$ independent trials, every one of which uses a different seed for the initial warmup design to ensure a fair comparison between different acquisition function performances.

The initial design occurs before training the GP models, where we run the model through a group of seeded randomized hyperparameter combinations and collect a set of observations, along with their corresponding objective values and stagewise runtime costs, to get enough data for model training, as well as to obtain an idea of the desired budget of each experiment. We collected $N_0 = 10$ warmup hyperparameter combinations, and set the total optimization budget as a function of the warmup budget. We ran all acquisition functions for both synthetic and real experiments on the same settings, including initial datasets used to train the GP models, in order to ensure a fair comparison. Every iteration, we generate $M = 512$ raw samples (potential candidate points) with $r = 10$ random restarts. We also sample $D = 1,000$ values from each of the cost and objective GP models, in order to estimate their corresponding expectations. Figures 3 and 4 omit progress during the warmup iterations, as said warmup exhibits similar results and behavior across all acquisition functions.

## 4.1 SYNTHETIC PIPELINES

For our synthetic experiments, we defined our own objective and cost functions, such that:

- The objective functions would have a complex analytical form.

- The cost functions would have a wide range of values spanning across the objective function domain, mainly to avoid uniformly distributed evaluation costs.

We define one synthetic BO function per stage, which takes the corresponding stage's hyperparameters as input variables, then define the objective as the sum of the stage-wise functions. Similarly, we define one cost function per stage as a combination of *cosine*, *sine*, *polynomial*, and *logistic* functions, while ensuring $c_k(x) \in \mathbb{R}^{+*}$ for every cost value across the objective function's search space. The objective functions the cost functions we designed are defined in Appendix C.

Figure 3 shows that, with the use of memoization, EEIPU manages to run for twice the iteration count of other acquisition functions using the same total budget, without compromising on the objective value achieved by the end of the optimization process. In Section 5, we discuss these results further, and quantify the performance of each acquisition function to highlight the superiority of integrating memoization with cost awareness.

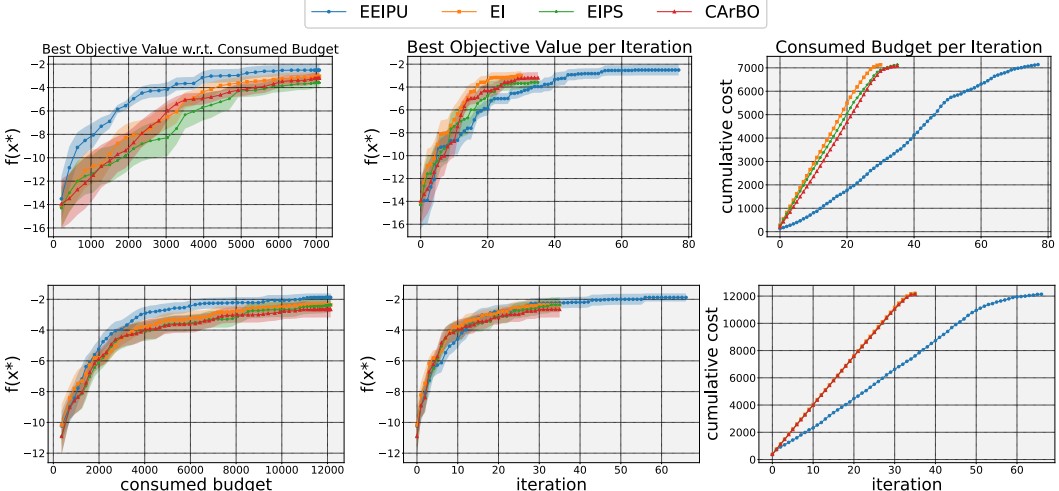

Figure 3: This figure shows every acquisition function's performance on both synthetic functions: $f(x) = \mathtt{Beale2D} + \mathtt{Hartmann3D} + \mathtt{Ackley3D}$ with stagewise cost functions $[3, 4, 5]$ per Table 5 (**Top Row**), and $f(x) = \mathtt{Branin2D} + \mathtt{Beale2D} + \mathtt{Michale2D}$ with stagewise cost functions $[1, 2, 3]$ (**Bottom Row**). Left plots show the best objective value achieved by each acquisition function with respect to the consumed budget, where a quicker progression to the top means a higher objective value achieved at only a fraction of the cost of other methods. Middle plots show the best objective value achieved with respect to the iteration count, which stops as soon as the allocated experiment budget is fully consumed. Right plots show the consumed cost per iteration, where a slower progress means a lower cost induced with every iteration.

## 4.2 SEGMENTATION AND STACKING PIPELINES

In this section, we define the two AI pipelines used in our real experiments, as summarized in Figure 4. We define the stage components, tunable hyperparameters, target output, as well as the evaluation metric (objective value). The stagewise cost of running these pipelines is defined by the wall time execution duration of the corresponding stage.

The stacking pipeline is a two-staged pipeline for classifying loan applicants into default or no default categories. The first stage fits an ensemble of four classifiers: *Extreme Gradient Boosting, Extra Trees Classifier, Random Forest*, and *CatBoost*, while the second stage fits a *Logistic Regression* classifier on the outputs of the first. For the first stage, we optimize six hyperparameters (that include number of estimators `C`, the maximum depth `max_depth`, and the learning rate `lr` across the four models), while for the second stage, we optimize three hyperparameters (regularization parameter `C`, tolerance parameter `tol`, and the maximum number of iterations `max_iter`). The objective used in this experiment is the area under the receiver operating characteristics *AUROC*.

The segmentation pipeline, on the other hand, is a pipeline for semantic segmentation of aerial images. It is a three-stage pipeline. The first stage is a data pre-processing stage with six hyperparameters. The hyperparameters are the mean $\mu$ and standard deviation $\sigma$ values for normalising the images across the RGB channels. The second stage is a *UNet* architecture for which we optimize the batch size `BS`, maximum learning rate `lr`, and weight decay. The final stage is a post-processing stage that applies a conditional random fields algorithm to suppress noisy pixels. We optimize the compatibility transform matrix `compat`, the tuple of standard deviation of the `x` and `y` dimensions of the segmentation mask `sdim`, and the probability that the ground truth probability is accurate `gt_prob`.

## 5 RESULTS

When looking at the best objective value achieved by every acquisition function with respect to the iteration count, we occasionally observe slower progress for EEIPU relative to the remaining acquisition function to converge towards a similar best objective value. However, with the help of memoized observations, EEIPU makes up for the slow progress with a much higher total number of iterations, eventually leading to a higher objective value achieved by the end of the optimization

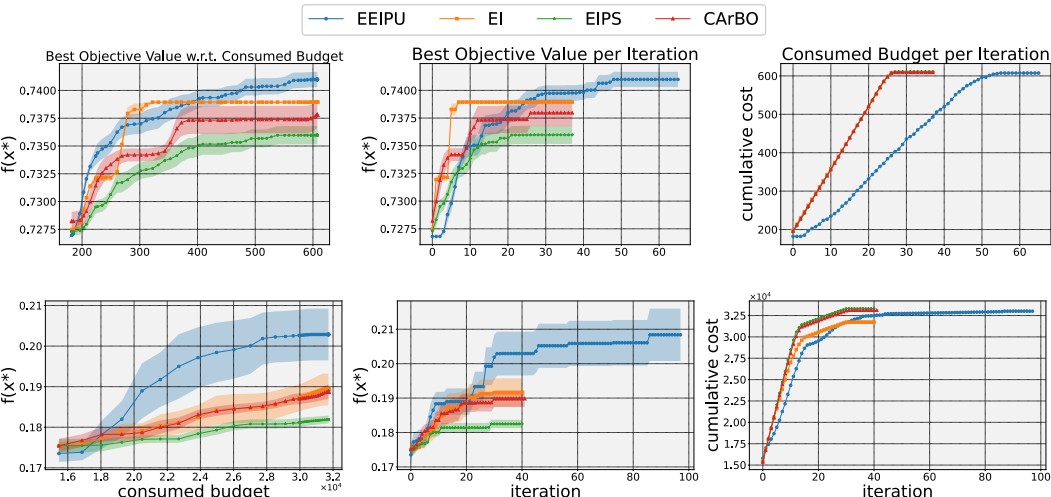

Figure 4: This figure shows summary plots of our real experiments. The stacking pipeline (**Top row**) is a two-stage pipeline with a total budget of 600 seconds, while the segmentation pipeline (**Bottom row**) is a three-stage pipeline with a total budget of 32,000 seconds.

process. Tables 2 quantifies and summarizes the performance of every acquisition function on both synthetic and real pipelines.

Table 2: Table summary of each acquisition function's performance on each pair of pipelines in our synthetic, as well as our real experiments. The functions are evaluated based on their average highest-achieved objective value by the time the total budget is fully exhausted in each trial.

|  | EEIPU | EI | EIPS | CArBO |
|---|---|---|---|---|
| Beale2D + Hartmann3D + Ackley3D | $\mathbf{-2.51 \pm 0.48}$ | $-3.04 \pm 0.39$ | $-3.59 \pm 0.46$ | $3.20 \pm 0.49$ |
| Branin2D + Beale2D + Michale2D | $\mathbf{-1.88 \pm 0.26}$ | $-2.36 \pm 0.37$ | $-2.36 \pm 0.48$ | $-2.65 \pm 0.53$ |
| Segmentation Pipeline | $\mathbf{0.741 \pm 1e^{-3}}$ | $0.740 \pm 1e^{-3}$ | $0.739 \pm 1e^{-3}$ | $0.738 \pm 1e^{-3}$ |
| Stacking Pipeline | $\mathbf{0.20 \pm 0.01}$ | $0.19 \pm 1e^{-3}$ | $0.18 \pm 1e^{-3}$ | $0.19 \pm 1e^{-3}$ |

EEIPU managed to outperform every comparable acquisition function, achieving state-of-the-art results within the allocated optimization budget, while managing a significantly higher number of iterations at every turn, as highlighted in Table 3 below.

Table 3: Table summary of each acquisition function's performance in our synthetic, as well as our real experiments. The functions are evaluated based on the number of iterations they completed during the optimization process before fully exhausting the allocated budget.

|  | EEIPU | EI | EIPS | CArBO |
|---|---|---|---|---|
| Beale2D + Hartmann3D + Ackley3D | **77** | 30 | 35 | 35 |
| Branin2D + Beale2D + Michale2D | **66** | 32 | 34 | 35 |
| Segmentation Pipeline | **65** | 36 | 37 | 37 |
| Stacking Pipeline | **97** | 39 | 41 | 41 |

## 6 CONCLUSION

In this paper, we demonstrated the potential of scaling cost-aware BO up to multi-stage pipeline optimization, highlighting the importance of introducing memoization into the optimization process. EEIPU has achieved state-of-the-art results on both synthetic as well as real experiments – it incurs significantly lower total cost per BO iteration compared to existing cost-aware methods. In particular, EEIPU memoizes earlier pipeline stages in order to reduce the cost of optimizing later stages; the result is fully-automated, pipeline-wide optimization at a fraction of the originally anticipated cost. As future work, we may consider studying certain limitations of EEIPU – as described in Appendix E – which arise when objective functions are strongly correlated with costs.

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

# APPENDIX

## A  PROBLEM SETUP

We consider a "black-box cost" multistage pipeline setting, in which the goal is to maximize a single objective function $f : \mathcal{X} \to \mathbb{R}$ with no gradients or analytical form. This function $f$ is the output of a $K$-stage AI pipeline, where $x_k \in \mathcal{X}_k$ are the tunable hyperparameters of stage $k$ of said pipeline. Let $x \in \mathcal{X} = \mathcal{X}_1 \times \mathcal{X}_2 \times \ldots \times \mathcal{X}_K$ be the set of hyperparameters across all stages, $\mathcal{Y} = f(\mathcal{X})$ be the objective value returned by the pipeline, and $\mathcal{C}_k$ be the cost of running stage $k$ of the $K$-stage pipeline. Observe that every stage $k$ has its own cost function, but there is only one objective function $f$ for the entire pipeline.

Under this setting, our goal is to find the optimizer of $f(x)$, while leveraging the properties of the multistage setting to minimize the total cost required over all iterations/queries. In this paper, we propose EEIPU, a cost-efficient memoization-aware BO acquisition function that, in a nutshell, evaluates the expected-expected improvement per unit cost of potential hyperparameter candidates $\mathcal{X}$. EEIPU uses uncertainty modeling to compute its value, and proposes a memoization-based caching mechanism to minimize the budget of running an observation through the pipeline, which increases the total number of iterations under a fixed optimization budget, therefore maximizing the chances of finding the optimizer of $f(x)$ compared to existing methods.

Our setting differs from that of LaMBO (Lin et al, 2021) and Multistage BO (Kusakawa et al, 2021), which are methods that implement a memoization-like technique for multi-stage pipelines., Importantly, neither method is applicable (at least, not without substantial modifications or assumptions) to our black-box cost setting. LaMBO employs a slowly moving bandit (SMB) algorithm that uses a tree structure to represent the tunable hyperparameters, partitioned into user-defined modules with the aim of minimizing the cost of updating hyperparameter modules between adjacent iterations. This approach relies on users to specify a partitioning of the hyperparameter space, something that is not required by EEIPU. While the LaMBO paper provides a "bisection" heuristic for partitioning the parameter space, this causes the tree size (number of leaves) to grow exponentially with increasing hyperparameter dimension. Furthermore, each LaMBO iteration requires solving a local BO problem for every leaf in the tree, hence limiting its usefulness in high-dimensional hyperparameter spaces.

As for Multi-stage BO, it requires stronger assumptions on cost functions than are admissible in our black-box cost setting: (1) their setting assumes prior knowledge of the cost of running every stage of their pipeline, and furthermore, they assume the tunable parameters do not affect costs; (2) Multi-stage BO requires each pipeline stage to have a separate objective function, whereas in our setting, there is only one objective function for the entire pipeline. Our setting is consistent with the majority of Machine Learning pipelines, where the desired objective - such as a validation accuracy or other metrics - is only known after the entire pipeline finishes execution; intermediate stages do not possess their own objective functions.

## B  ADDITIONAL EXPERIMENTS

### B.1  SYNTHETIC EXPERIMENT

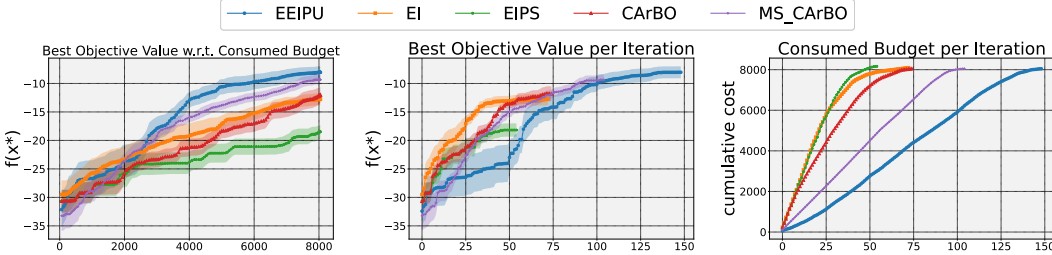

Figure 5: This figure shows the performance of all acquisition functions on a 30-dimensional 5-stage pipeline with a budget of $8,000$ cost units.

## B.2    REAL EXPERIMENTS

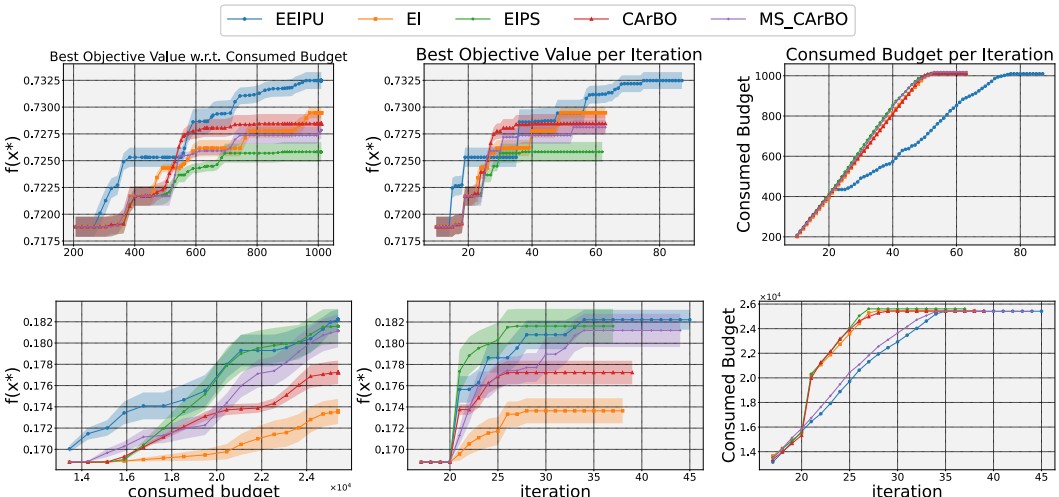

Figure 6: This figure shows summary plots of our real experiments. The stacking pipeline (**Top row**) is a two-stage pipeline with a total budget of $1,000$ seconds, while the T5 Distillation pipeline (**Bottom row**) is a three-stage pipeline with a total budget of $25,000$ seconds.

## C    SYNTHETIC FUNCTIONS

The synthetic functions we used in our experiments, detailed in Section 4.1, are standard BO functions defined in Table 4 below.

Table 4: Objective Functions used for Synthetic Pipelines

| Function Name | Formula |
|---|---|
| Branin-2D (Dixon & Szegö, 1978) | $(x_2 - \frac{5.1}{4\pi^2}x_2^2 + \frac{5}{\pi}x_1 - 6)^2 + 10(1 - \frac{1}{8\pi})\cos(x_1) + 10$ |
| Hartmann-3D (Hartmann, 1972) | $-\sum_{i=1}^{4} \alpha_i \exp(-\sum_{j=1}^{3} A_{ij}(x_j - P_{ij})^2)$ |
| Beale-2D (Nocedal & Wright, 2006) | $(1.5 - x_1 + x_1x_2)^2 + (2.25 - x_1 + x_1x_2^2)^2 + (2.625 - x_1 + x_1x_2^3)^2$ |
| Ackley-3D (Ackley, 1987) | $-20\exp(-0.2\sqrt{\frac{1}{3}\sum_{i=1}^{3}x_i^2}) - \exp(\frac{1}{3}\sum_{i=1}^{3}\cos(2\pi x_i)) + 20 + e^1$ |
| Michale-2D (Michalewicz, 1996) | $\sum_{i=1}^{2}\sin(x_i)(\sin(ix_i^2/\pi)^{20})$ |

Table 5: Synthetic Functions Used for Evaluating Stage Costs

| Cost Type | Function | Stage Dimension | Formula |
|---|---|---|---|
| 1 | Logistic + Cosine | 2 | $20\cos(x_1) + \frac{100}{1+\exp(-5x_2)} + 60$ |
| 2 | Logistic + Polynomial | 2 | $\frac{20}{1+\exp(-3x_1)} + x_2^3 + 100$ |
| 3 | Cosine - Sine | 2 | $50\cos(x_1) - 20\sin(x_2) + 100$ |
| 4 | Polynomial + Cosine + Sine | 3 | $5x_1^2 + 30\cos(x_2) + 15\sin(x_3) + 50$ |
| 4 | Logistic + Cosine + Polynomial | 3 | $\frac{20}{1+\exp(-4x_1)} + 30\cos(x_2) + x_3^3 + 75$ |

## D    EEIPU'S THEORETICAL UPPER-BOUND

EEIPU's theoretical upper-bound, referred to as EIPU-MEMO, is an oracle version of our method which has full access to the true cost function, therefore bypassing the need for fitting stagewise cost GPs. While EEIPU is theoretically able to achieve a higher objective value due to the randomness that stems from modeling costs, EIPU-MEMO is guaranteed to use the true cost of an observation to compute EIPU$(X)$. Therefore, we consider EIPU-MEMO to be an oracle version of EEIPU which behaves exactly as intended by our algorithm. Figure 7 shows the behavior of EEIPU and EIPU-MEMO when ran on our synthetic pipelines.

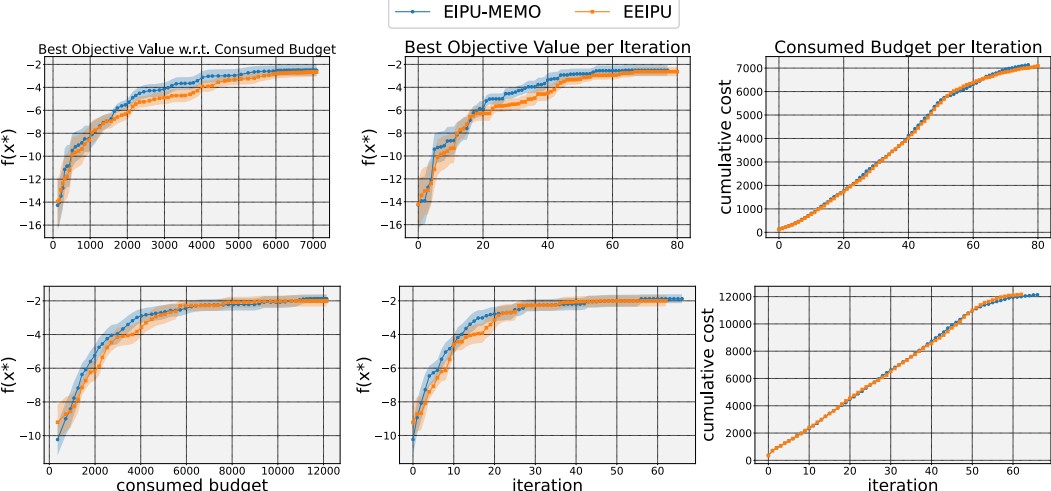

Figure 7: This figure shows summary plots of EIPU-MEMO's performance on our synthetic pipelines compared to EEIPU.

EIPU-MEMO follows the intended behavior of our cost-aware design, which leads it to finding better objective values at a slightly faster pace than EEIPU. However, the consumed budget per iteration (right) plot shows that EEIPU's cost modeling process succeeds in approximating the majority of the costs chosen by EIPU-MEMO, which is a good indicator of EEIPU's ability to model and estimate costs.

## E  LIMITATIONS

When the objective function we are optimizing for is positively correlated with the cost function, the optimal objective value $f(x^*)$ would be located at a very high-cost region. In this case, sample-efficient techniques, such as EI (Mockus et al., 1978) could potentially find the optimal objective value by immediately targeting the high cost regions where the expected improvement is maximized, while EEIPU is initially directed towards low-cost regions for a large part of the optimization process.

To overcome this limitation, we speculate that the following procedure may be effective: the warmup iterations can be altered to prioritize hyperparameter choices in high-variance/uncertain regions of the search space. Intuitively, this variation of EEIPU initially prioritizes the acquisition of cost and objective information over a wide search space. When objective is correlated with cost, this allows high-cost-high-objective regions to be quickly discovered in earlier iterations, without wasting iterations on low-cost-low-objective regions. Future work may also include early stopping mechanisms where the optimization process is stopped (based on criteria to be developed) before the cost budget is fully exhausted.

## F  DETAILS OF THE COMPUTE ENVIRONMENT

We set up our experiments on a server node with 4 NVIDIA A100-SXM4-40GB GPUs and 128 CPU cores with 2 threads per core. While our node had 4 GPUs, our experiments only used 1 GPU at a time, since parallelization was unnecessary for the data and model sizes in our pipeline experiments. For more resource-intensive pipelines, any of the commonly-used parallelization methods (e.g. data-parallelism, model-parallelism or pipeline-parallelism) can be applied to the pipeline; this would not interfere with EEIPU's proper functioning.

