# OpenReview forum: "Memoization-Aware Bayesian Optimization for AI Pipelines with Unknown Costs"
_ICLR.cc/2024/Conference — Submitted to ICLR 2024_

### Official Review · Reviewer_fSnX · 2023-10-26

**Soundness:** 3 good
**Presentation:** 3 good
**Contribution:** 1 poor
**Rating:** 5
**Confidence:** 4

**Summary:**

This paper generalizes the cost-aware Bayesian optimization (BO) algorithm from single-stage to multi-stage optimization problem. To achieve this, a new Expected-Expected Improvement Per Unit-cost (EEIPU) acquisition function is proposed, and a memoization-awareness trick is considered for improving the cost efficiency. Empirical results on both synthetic and real experiments show that the proposed EEIPU outperforms conventional EI and cost-aware BO baselines.

**Strengths:**

- The problem in this paper is well motivated and the writing style is good. The proposed algorithm and main idea of this work are easy to follow.

- The experimental design is valid. Both synthetic functions and real-world AI pipelines are used to test the proposed algorithm.

**Weaknesses:**

- The novelty of the proposed method is limited. The proposed method is a combination of cost-aware BO, unknown-cost modeling, cost cooling, and the memoization trick. All these techniques are not new and have been used in existing BO works (as shown in Table 1). The technical issues or challenges of combining them are not clearly shown in this work. The proposed EEIPU simply replaced the cost function $c(x)$ of EIPU with a total cost function $C(x)$ which is defined as the sum of $c(x)$ over multiple stages. Although the author(s) claimed that computing the expected total inverse cost $\mathbb{E}[1/C(x)]$ is not a straightforward task, the issue is resolved by conventional MC sampling which does not contribute to a non-trivial solution.

- The important baselines are not fairly compared or discussed. In Table 1, why Multi-stage BO is labeled as not cost-aware or memoization-aware? The "stock and resume" scheme of Multi-stage BO is a very similar concept to the "memoization" here and the cost has been considered in Algorithm 2 of (Kusakawa et al., 2021). As shown in Table 1, LaMBO and Multi-stage BO should be the most related works which, unfortunately, are not compared in the experiments. The difficulties of reproducing LaMBO are discussed in Section 3.4. However, it's not clear why Multi-stage BO is not tested. Also, even if both algorithms are hard to reproduce, a detailed discussion about the novelty of EEIPU compared to these two baselines is needed for showing the significance of this work. In particular, since the known and fixed cost are claimed to be the major issues of Multi-stage BO and LaMBO (Section 2), is there any difficulties in generalizing their algorithms to the unknown-cost setting? Compared to the memoization strategies shown in these two works, what is the superiority of the proposed memoization method? Are there any scenarios that can be tackled by EEIPU instead of Multi-stage BO or LaMBO?

**Questions:**

- At the end of Section 2, it is claimed that "In our setting, the cost at a given stage is a function of the hyperparameters provided to that stage, as well as the inputs provided from the previous stage". Can you provide more technical details or examples to support this sentence? What are the "inputs from the previous stage"? Why and how are these inputs used to model the cost function? Is it shown in any part of Section 3?

- What's the value of $\epsilon$ used in the experiments? Is the experimental results sensitive to the settings of $\epsilon$, $M$, or $N$?

- In Fig. 3&4, why does EI outperform both EIPS and CArBO? It seems to be counter-intuitive and inconsistent with the results reported in existing works. The y-axis labels of the graphs in the middle column of Figs. 3&4 seem to be wrong. Shouldn't it be the function value instead of the stage cost? The right graphs of Fig. 3&4 each have one plot that the cumulative costs of EI, EIPS, and CarBO are almost the same. Can you provide some analyses on this observation?

---

> ### Author Response · Authors · 2023-11-23
> **Part 1**
>
> Thank you for your detailed review of our paper, as well as the observations you made, all of which will be addressed below. First, we would like to direct you to the global rebuttal section, as well as the new appendix of our draft, for a summary of all our rebuttal efforts to clarify our proposal.
>
> ### **Q1: Novelty**
>
> **A1**: While the components of EEIPU, such as black-box cost modeling and cost-cooling, are inspired by existing cost-aware techniques, there is no method that  generalizes cost-awareness to the multi-stage setting, which our results show reduces costs using memoization even in cases where existing cost-aware methods fail to do so. Existing memoization techniques also differ from our setting in several ways, which we elaborate on below:
> - LaMBO (Lin et al, 2021) is a technique that aims to minimize the switching costs from one hyperparameter combination to another across two consecutive iterations. They model the hyperparameter search space as a multi-armed bandit, choosing a suffix of stages to update every iteration with the aim of minimizing the number of updated stage parameters from the previous iteration. This method, however, has no knowledge of the cost function of each space, and makes no use of it during the optimization process.
> - The Multistage BO (Kusakawa et al, 2021) paper explores the potential of multi-stage pipeline optimization in an industrial setting rather than a theoretical one. The scope of this paper covers pipelines with stagewise intermediate outputs, each of which is modeled by a separate GP, which aids in suspending the pipeline at a stage $k$ if a stage’s output is too far off its modeled expectation. This allows the storing of the $K-1$ first stages, to be used by future iterations for a potentially better final objective than the suspended process. This technique implements a memoization variant, but makes no use of the costs incurred by the corresponding stages.
> - While these techniques introduce a memoization-like mechanism, EEIPU is the first method to incorporate memoization-based cost efficiency into the querying process. Using uncertainty modeling, EEIPU is not only able to maximize the exploitation of cheap-to-evaluate areas of the search space through cost awareness, but is also able to use memoized observations to explore high-cost regions without having to incur the cost of rerunning most of the pipeline stages for a second time. This combination allows for faster (less costly) exploration of the search space. Thus, with EEIPU we observe a much higher number of evaluations compared to other methods, when all methods are given  a similar budget. Considering that almost all BO methods are able to reach the global optimum as the number of iterations increases, EEIPU is more likely to reach a higher objective value under budget constraints.
>
> ### **Q2: Why not use simple MC Sampling to compute expected inverse cost?**
>
> **A2**: Due to the nature of our stagewise cost-modeling and discounting strategy, we are unable to rely on MC sampling alone to model the expectation of the total inverse cost, as explained in section 3.1 of our paper. However, we acknowledge that our current approach may not be the most accurate way to represent this expectation, and we are actively working towards a more theoretically grounded technique, such as a Taylor expansion.
>
> ### **Q3: Typos**
>
> **A3**: We apologize for the typos in Table 1, Figures 3 and 4, and the last sentence of the Background, all of which have been rectified.

---

> ### Author Response · Authors · 2023-11-23
> **Part 2**
>
> ### **Q4: Why not adapt LaMBO and MS_BO to your setting?**
>
> **A4**: Regarding LaMBO and Multistage BO, the brief explanation is that both methods require additional information that are not available in the pipeline setting that EEIPU is designed for. In contrast, EIPS and CArBO do not require these additional pieces of information, and are runnable in the same setting as EEIPU. We explain further:
> - LaMBO employs a slowly moving bandit (SMB) algorithm, utilizing a tree structure where each arm represents a candidate variable or hyperparameter in a tuning case. The cost of switching between arms is encoded in the tree, and the switching cost is defined as the distance between the leaves corresponding to the two arms. Importantly, for LaMBO to construct its tree, it needs to be provided with a user-constructed division of the hyperparameter space into disjoint partitions.  In Section 3.3 of the LaMBO paper, the authors propose a simple “bisection” heuristic for creating these partitions, but this causes the number of LaMBO tree leaves to grow exponentially with the hyperparameter dimension, in turn causing LaMBO’s computational requirements to also grow exponentially (as it requires evaluating a local BO problem at each leaf). This makes LaMBO not as well-suited to high-dimensional problems. We had reached out to the LaMBO authors to request code and start a discussion, but did not receive a response. Due to these difficulties, we chose to prioritize baselines that could scale well with higher dimensions, such as CArBO and its multi-stage variation. We regret that we could not compare against LaMBO in this paper submission, owing to these circumstances. We hope to provide a LaMBO benchmark in a future work, as it will require substantial effort and time to reproduce (and validate that the reproduction is indeed correct) absent any author-provided code.
> - We cited Multistage BO for its stock and resume mechanism that most resembles our memoization technique. There are two key differences from our EEIPU setting: (1)  Multistage BO requires stronger assumptions on cost functions.- They assume an easier setting with prior knowledge of the cost of running every stage of their pipeline, and  the tunable parameters do not affect costs at all. Modeling of the cost function, as EEIPU does, is unneeded in this easier setting. (2) Multi-stage BO requires each pipeline stage to have a separate objective function, whereas in EEIPU’s setting, there is only one objective function for the entire pipeline. To run Multistage BO in our setting, we would need to make additional strong assumptions, such as assuming that every stage has the same objective function.  As a point of clarification, in our EEIPU setting, stage outputs are only used for memoization; we do not assume each stage (except the final one) produces an  objective value that can be optimized for. This assumption is consistent with most Machine Learning pipelines, where the desired objective to be optimized, such as a validation accuracy or other metric, is only known after the entire pipeline finishes execution.
>
> For these reasons, we hope we have clarified why the unimplemented techniques either could not be generalized to our setting, or presented unusual difficulties. We also hope that the new and improved experiments we ran in this rebuttal phase, summarized in the global rebuttal section, could portray our efforts to provide a fair and comprehensive assessment of our approach.
>
> ### **Q5: Is performance sensitive to $\epsilon, M,$ and $N$?**
>
> **A5**: $\epsilon$ is only set in our synthetic experiments and has no impact on performance. $M$ and $N$ are parameters used by all BO techniques in the optimization process. We set them in our experiments to their default values, whereas users can tune them freely.
>
> ### **Q6: What causes EIPS and CArBO to perform poorly sometimes?**
>
> **A6**: The relatively poor EIPS and CArBO performance compared to EI on the stacking pipeline is further analyzed in the global rebuttal section.
>
> We would like to thank you again for your valuable review, and apologize for the lengthy reply we drafted. However, we hope that our detailed analysis of the existing methods provided you with a clearer view on our novelty and understanding of our chosen baselines.

---

### Official Review · Reviewer_uCj5 · 2023-10-30

**Soundness:** 3 good
**Presentation:** 3 good
**Contribution:** 2 fair
**Rating:** 5
**Confidence:** 3

**Summary:**

The authors propose Expected-Expected Improvement Per Unit Cost (EEIPU), a cost-aware BO algorithm able to handle multi-stage pipelines with independent, unknown costs. The acquisition is computed via Monte Carlo sampling to estimate the expected inverse cost which are modelled using GPs on log costs. The multi-stage pipeline is exploited via memoization of previously observed candidates and intermediate outputs. When selecting new candidates, some new candidates reuse the memoized results to avoid incurring costs computing the memoized results again. EEIPU is empirically evaluated with comparison against suitable baselines.

**Strengths:**

1. The paper is generally written clearly and is easy to understand.
2. EEIPU empirically outperforms previous algorithms when the assumption of a multi-stage pipeline holds.
3. The experiments section is well-designed, the segmentation and stacking pipelines are realistic.

**Weaknesses:**

1. The assumption of a multi-stage BO process with independent stages and observable intermediates is a very strong one that can be exploited more than this work currently does. This work only exploits this structure to memoize previously evaluated candidates to save costs expended during the BO process. This structure has been exploited to improve the modelling and guide the search, for example, see "Bayesian Optimization of Composite Functions" (Astudillo and Frazier, 2019) and "Bayesian Optimization of Function Networks" (Astudillo and Frazier, 2021).

2. Perhaps the $M$ prefixes can be sampled more intelligently than randomly, e.g., by weighting the prefixes based on the results of previous evaluations incorporating those prefixes.

3. The memoization method fails if the intermediate stage outputs are noisy. While AI pipelines could be assumed to have noiseless stages, it is not inconceivable that these stages are noisy, e.g., if the stages are generative models.

4. Some clarifications, see Questions section, in particular Question 2 about the experimental results.

5. Typos/inconsistencies: 1) Written as $EI \times \mathbb I^\eta$ in Algorithm 1, but $EI * \mathbb I$ and $EI * \mathbb I^\eta$ in Equations (5) and (6); 2) in Figures 3 and 4, left and center column plots are supposed to be best objective value, but the y-axis is written as $f(x^*)$ in the left plots and 'stage costs' in the center plots; 3) [()] described as 'empty prefix' and 'empty subset' in different parts of Sec. 3.2.

**Questions:**

1. What is the rationale of introducing the $\epsilon$-cost for stages 1 to $\delta$? Why not set to $0$?

2. There are a few peculiarities with the results for EIPS and CArBO in Figures 3 and 4. In Figure 3 bottom row right plot and Figure 4 top row right plot, EIPS and CArBO consume the exact same cost in each iteration as the non-cost aware EI. This is very strange since EIPS and CArBO are supposed to be cost-aware. In addition, in all the results shown, the non-cost aware EI outperforms EIPS and CArBO in terms of best objective value achieved against cost incurred, which is again strange given that EIPS and CArBO were designed for this very setting. Could you investigate and explain these anomalies?

---

> ### Author Response · Authors · 2023-11-23
>
> Thank you for the in-depth comments on our paper, and the valuable insights you have provided. To answer your question about the performance of CArBO and EIPS compared to EI, we would like to refer you to our global rebuttal section, where we introduce a new acquisition function, MS_CArBO, that is designed to help understand the true power of memoization, in the events where hyperparameters have little to no impact on the cost function. Additional clarifications and experiments are also summarized in the new appendix of our draft. We will now respond to each of the points you raised in your review.
>
> ### **Q1: The setting of independent stages and intermediate observations can be exploited more. There may be a smarter way to sample the $M$ candidates**
> **A1**: The papers you referenced provide a greater view on the potentials of multi-GP modeling approaches, and suggest potentially valuable expansions to our method.
>
> Our cost modeling strategy could especially benefit from the setting explored by the paper “Bayesian Optimization of Function Networks” (Astudillo and Frazier, 2021), their concept could be applied to our cost GPs for identifying a number of stages to run in order to minimize costs, while conditioning this process on the search space of the corresponding objective value, which could further maximize the EEIPU value.
>
> The current scope of this paper serves as a proof of concept, dedicated to showcasing the potential of a memoization-based approach in the cost-aware BO setting. Further improvements are being explored for EEIPU, including your suggestion to implement a smart sampling technique, following this initial proposal. Thank you for making the valuable observation.
>
> ### **Q2: Memoization fails when encountering a stage with noisy outputs**
> **A2**: We acknowledge that our method is not applicable to pipelines which require a true (i.e. non-pseudorandom) or external source of randomness. Our setting considers AI applications where pseudorandom number generators are given fixed seeds, which is a common approach in research and production AI because it ensures reproducibility of results, and makes debugging easier.
>
> ### **Q3: Why set costs to $\epsilon$ instead of zero?**
> **A3**: We set the cost of running memoized stages to $\epsilon$, because there is always an overhead cost of retrieving outputs from the cache, which can never be zero.
>
> ### **Q4: Why do EIPS and CArBO have similar cost to EI in some cases?**
> **A4**: This is a nuanced issue and we would like to explain as best we can. Our experiments show two differences in the behavior of EIPS/CArBO vs EEIPU, which explain why multi-stage cost GPs (as seen in EEIPU but not the other baselines) are important for the pipeline setting: (1) EIPS/CArBO use a single GP to model the cost of the entire pipeline, but for some (though not all) of our synthetic and real experiments, this single-cost-GP assumption turns out to be too coarse-grained: EIPS/CArBO are unable to discover low-cost regions in these multi-stage pipelines; rather, they chose hyperparameters whose cost only differs very slightly from EI (we apologize that this is difficult to see on the Figures; we’ve doubled checked our numbers and confirm the EIPS/CArBO costs are not identical to EI). (2) Even for experiments where EIPS/CarBO incur visibly different cumulative costs versus EI, our proposed EEIPU still has lower cumulative costs. This is due to the substantial cost savings from re-using memoized results. We also note that EEIPU's use of stage-wise cost GPs is a prerequisite for effective memoization, because we cannot model the cost of partially executing a pipeline with the EIPS/CarBO single-cost-GP approach. We will put this discussion into the paper if it is accepted.
>
> ### **Q5: Typos**
> **A5**:
>
> - For the EEIPU formula: In section 3.2, we described our cost-discounting process, before showing how the mechanism contributes to EEIPU. The $\eta$ factor was not yet introduced, and was kept for section 3.3, where we defined cost-cooling, and proceeded to include $\eta$ in the equation.
> - Thank you for pointing out the typos in our plot labels. Since we commonly use $f(x*)$ to denote the best objective value, we have updated our plots to display it.
> - In our paper, we refer to the empty prefix by an empty list [()]. However, when referred to as a subset, we mean the subset {[()]}, which is part of the global set of stored prefixes $P_X$. We apologize for that oversight.
>
> Thank you again for your observations, all of which have been rectified in the new draft, and some of which have provided us with meaningful insights on possible expansions of our method. We hope that our replies, as well as the expansion to our appendix, have provided enough clarity to establish the merit of our paper.

---

> > ### Comment · Reviewer_uCj5 · 2023-12-05
> >
> > Thank you for your response. I am keeping my score unchanged as I believe this work has the potential to be much better than its current form. My main concern remains that the very strong assumption of a multi-stage pipeline is not exploited enough, and other reviewers have also commented on the simplicity of the approach. There is nothing wrong with simple ideas, but when they require very strong assumptions, the work becomes less impressive. I believe there is space for this work to be greatly improved, and I hope that this review process has been helpful towards that goal.

---

### Official Review · Reviewer_gsaT · 2023-10-30

**Soundness:** 3 good
**Presentation:** 3 good
**Contribution:** 2 fair
**Rating:** 6
**Confidence:** 4

**Summary:**

This paper addresses the optimization of hyperparameters for a multi-stage AI pipelines with unknown costs.  By utilizing Bayesian optimization, it solves a black-box optimization problem on the AI pipelines.  Since each stage depends on the previous stages and its computational cost varies, it needs to separately model final function evaluations and costs.  Notably, it extends the expected improvement acquisition function to the expected-expected improvement per unit cost, by calculating the expected inverse costs.  Eventually, the authors show some experimental results on several benchmarks with multiple stages.

**Strengths:**

* It solves an interesting problem related to multi-stage AI pipelines, defined by considering practical scenarios.
* Proposed method is well-motivated.
* Paper is generally well-written.

**Weaknesses:**

* More compelling experiments can be conducted.  I think that the experiments tested in this paper seem interesting, but the scale of experiments is small compared to the common scale of the experiments in ICLR.
* I don't fully agree with the need of memoization.  Are the processes memoized really computationally expensive?  I think that the numerical analysis can be provided in order to strengthen the motivation of memoization awareness.

**Questions:**

* The reference by Mockus (1998) might be published in 1978, not 1998.
* Why should costs be positive?  Is it a mandatory condition?
* In Figure 2, why are $c_1$ and $c_2$ zero in (c) and (d)? Are they correct?  If they are correct, please add description in the rebuttal.
* Why are the best objective values at iteration 0 are different across methods?  Initialization should be identical across tested methods, such that the initial values should be same.
* The captions of tables should be located above the tables.
* I think that the authors need to update Figure 1.  Fonts for mathematical expressions are different from ones in the main article.

---

> ### Author Response · Authors · 2023-11-23
>
> Thank you for your valuable review, which directed our attention to rectifying the details you pointed out, including the EI reference, which we found to be indeed originally published in 1978, the captions on top of the tables, and the mathematical expressions in Figure 1. We have implemented a new acquisition function baseline, MS_CArBO, to further establish the merit and practical advantages of memoization. We also added two different pipelines (1 real and 1 synthetic) as additional sets of experiments, all of which are included in the appendix of our paper (to be moved up to the main content should we be accepted). Please allow us to expand on your points below, one at a time.
>
> ### **Q1: More compelling experiments can be conducted**
> **A1**: We designed  experiments that showcase a proof of concept on the synthetic pipelines, before testing practicality by optimizing hyperparameters for heavy models, in order to show the advantages of memoization-discounted costs in a practical setting. However, we understand the need for more in-depth experimental analysis, and have run two additional experiments for that reason.
> #### **Synthetic**
> We ran a high-dimensional synthetic experiment, where the objective-GP fits an ensemble of 30 hyperparameters against the final objective value, and 5 different cost-GPs are trained, with a potential of memoizing up to 4 stages of the pipeline.
> #### **Real**
> This experiment optimizes a model consisting of a three-staged T5 transformer fine-tuning and distillation pipeline whose first stage is data preprocessing and has a single parameter, batch size. The second stage is a fine-tuning stage with three hyperparameters, namely learning rate, weight decay and number of epochs. This stage fine-tunes a pre-trained Google t5-small model with 60 million parameters. The final stage is a distillation step with four hyperparameters – learning rate, temperature, weight decay and number of epochs. This stage distills the 60 million-parameter model to a 7 million-parameter model.
>
> The results of this experiment are plotted in the new appendix of our paper, and are also displayed on a table in the global rebuttal section.
>
> ### **Q2: I don’t fully agree with the use of memoization**
> **A2**: During this rebuttal period, we have dedicated a portion of our experiments to showing the true impact of memoization in discounting costs, especially when hyperparameters have little to no impact on the runtime costs. We designed a new acquisition function, referred to as MS_CArBO, it is a multi-stage version of the CArBO method with stagewise cost-GPs with no use of memoization, which most accurately portrays the advantage of memoization in reducing evaluation costs, therefore leading to a higher number of iterations. In our synthetic experiment, EEIPU manages to run for 44 iterations more than MS_CArBO, which led to a higher objective value. In the real experiment, EEIPU ran for 24 more iterations than MS_CArBO in the stacking pipeline where hyperparameters have no impact on the costs, and 1 extra iterations in the T5 distillation pipeline. These results, while not consistently staggering in difference, prove a degree of robustness in our proposal in challenging settings where existing BO approaches continue to fail.
>
> ### **Q3: Why should costs always be positive?**
> **A3**: In the context of our paper, cost-GPs are designed to model the cost of running a parameter combination through an AI pipeline, which typically refers to the wall-clock time, but could also be set as GPU usage, energy consumption… We, as well as existing cost-aware BO techniques, assume all measures of incurred costs to be positive, hence the modeling of log cost.
>
> ### **Q4: Figure 2 Ambiguities**
> **A4**: In Figure 2, we introduce memoization by example, and communicate the idea of discounted costs by showing how the cost of running a parameter combination through a stage of the pipeline goes from a certain value the first time it appears, down to zero when chosen again as a candidate by the memoization algorithm. In reality, there is always a negligible cost incurred even if we retrieve stage outputs from the cache memory. Therefore, memoization costs would always have to take an epsilon value, but that is not currently shown in this introductory Figure. We will update the Figure to show the epsilon term.
>
>
> ### **Q5: Different $f(x_0)$ for Different Acquisition Functions**
> **A5**: The initial data is common across all acquisition functions, which leads to a common best objective value at the start of the optimization process. However, our logging process starts at the end of  iteration 0, which increases the chances of different acquisition functions finding a better objective value in the first iteration.
>
> We would like to thank you again for your valuable review, and we hope that our response and additional experiments have successfully addressed your concerns.

---

### Official Review · Reviewer_W9CP · 2023-11-01

**Soundness:** 3 good
**Presentation:** 3 good
**Contribution:** 2 fair
**Rating:** 5
**Confidence:** 3

**Summary:**

This paper presents a method for cost-aware multi-stage BO that is also memoization-aware, meaning that the method stores the outputs of intermediate stages so as to reuse later. This allows for the proposed method to perform more function evaluation compared to other methods without memoization-awareness given the same budget.

The paper proposes a new acquisition function for this purpose, namely Expected-Expected Improvement Per Unit-cost (EEIPU). The EEIPU consists of 3 components:
- Cost awareness: the method assumes the cost is unknown, so they will build k GP models for k stages to represent the cost, then compute the expected cost by random sampling with Monter Carlo simulation.
- Memoization awareness: the proposed method stores the cost and output for each previous k-1 stages. Then, depending on the number of stored stages, the cost is discounted (but keep some overhead as a small cost), reducing the overall cost.
- Cost cooling: due to the nature of EEIPU, the proposed method may prioritize low-cost regions throughout the optimization, the author proposes to apply a cooling process (seems to base on the paper Lee et al., 2020) for the cost computation. The idea is to apply an exponential decay factor η∈[0,1], which gradually decreases with every iteration, so that eventually, the EEIPU turns back into the common EI acquisition function.

**Strengths:**

-	The idea of memoization is nice, it can be applied to increase the number of function evaluations for more knowledge to feed into BO process given a fixed budget.
-	The method may do well in the scenario that the budget is the running time allowed for optimization process. This has been proven by the experimental results (left columns of Figure 3,4), where given the same budget, EEIPU can obtain better outputs.
-	The paper writing is clear and easy to understand (although there is one thing regarding the problem setting I will mention in the Weaknesses section).

**Weaknesses:**

- As the topic covered in this paper is quite new, it will aid more if the paper includes a problem statement to describe in detail the problem setting. It took me quite some efforts to go back and forth to understand the setup of the problem tackled in this paper.
-	Please correct me if I'm wrong but it seems the application for this memoization technique is limited, as the stored data in the previous stages can only be reused when a repeated input for a certain stage is queried again (this is related to the 2nd question in the Question section).
-	The number of benchmarks is too few: only 2 synthetic and 2 real-world benchmarks, and with low dimensions.
-	In the synthetic benchmarks, it seems the improvement is only because the method manages to evaluate more data. Given the same number of iterations (middle column of Figure 3), even methods without cost-awareness like EI can find similar (or even some better) results. So this seems to me that the memoization only helps with increasing the evaluated data, but not really help much with the modelling of the surrogate model or the BO process.
-	The novelty of the work seems to be a bit limited. The idea of memoization and how it is incorporated in the proposed method seem to be a bit simple, while the effectiveness of the memoization to help with the modelling of the surrogate model or the BO process seems to be not clear. There is no deep insights to justify the proposed techniques. Finally, the cost cooling process seems to just inspire from previous works without modification.

**Questions:**

Besides answering my comments in the Weaknesses section, the authors could answer my following questions:
- What are the specifications of GPs using for the cost modelling? Which priors, kernels and hyper-parameter settings are used?
- For synthetic benchmarks, what is the input space for each stage? Is it discrete? If it is continuous, I'm just wondering how exactly does EEIPU re-use the stored data? It is hard for the acquisition function to propose the exact same points in the continuous domain. From the right column of Figure 3, it seems that the amount of reusing data is approximately 40 times over a total of 80 function evaluations.

---

> ### Author Response · Authors · 2023-11-23
>
> Thank you for the great amount of attention and analysis put into understanding our proposal, as well as valid suggestions for improving our paper. We have defined our problem setting, as well as the uniqueness of our proposal compared to existing memoization-based approaches in a new Problem Setup section. We have also carried out a number of new experiments (while we are unable to fully reformat the paper for this rebuttal due to the new experiments that will require moving some plots to the appendix, we have included all of our efforts in the appendix section for the this rebuttal).
>
> ### **Q1: The number of benchmarks is too few**
> **A1**: Our synthetic experiments were designed as a proof of concept of EEIPU, before testing its practicality by optimizing hyperparameters for heavy models, in order to show the advantages of memoization-discounted costs in a practical setting. However, we understand the need for more in-depth experimental analysis, and have run two additional experiments for that reason, described below.
> #### **Synthetic**
> We ran a high-dimensional synthetic experiment, where the objective-GP fits an ensemble of 30 hyperparameters against the final objective value, and 5 different cost-GPs are trained, with a potential of memoizing up to 4 stages of the pipeline.
> #### **Real**
> This experiment optimizes a model consisting of a three-staged T5 transformer fine-tuning and distillation pipeline whose first stage is data preprocessing and has a single parameter, batch size. The second stage is a fine-tuning stage with three hyperparameters, namely learning rate, weight decay and number of epochs. This stage fine-tunes a pre-trained Google t5-small model with 60 million parameters. The final stage is a distillation step with four hyperparameters – learning rate, temperature, weight decay and number of epochs. This stage distills the 60 million-parameter model to a 7 million-parameter model.
>
> The results of this experiment are plotted in the new appendix of our paper, and are also displayed on a table in the global rebuttal section.
>
> ### **Q2: Memoization needs to re-query the same inputs to be useful**
> **A2**: Memoization is indeed only useful when re-querying an input from a previous iteration. In section 3.2 of our paper, we describe how we guarantee the resampling of previously queried inputs by fixing some candidates’ prefix values from previous evaluations, while randomizing the values of the remaining parameters. This approach guarantees the inclusion of re-queried inputs.
>
> ### **Q3: EEIPU is only useful because it runs for more iterations**
> **A3**: Bayesian Optimization guarantees the optimization of the search space with high sample efficiency. Given enough iterations, most BO techniques would be able to reach a similar objective value. Our method leverages this ability, and further improves it by increasing the cost efficiency per evaluation, leading to a higher number of iterations under the same budget, therefore increasing our chances of achieving a higher objective value compared to existing methods.
>
> ### **Q4: The novelty seems to be limited**
> **A4**: The idea of memoization is indeed the main novelty of our proposal, whereas cost-cooling is an existing approach to motivate the exploration of high-cost areas. While caching techniques are common and explored in the context of BO, EEIPU is also the first method that supports black-box costs in a multistage setting through multi-GP modeling. By using memoization to minimize the cost of every iteration, our method not only allows the sampling of cheap observations, but also reduces the cost of previously expensive observations with potentially high objective values, by using cache-retrieved outputs to avoid incurring the costs of their corresponding prefix stages. EEIPU is therefore more likely to evaluate more areas of the search space, including expensive ones, than any existing BO technique in the multistage pipeline setting.
>
> ### **Q5: What are the specifications of the GP models?**
>
> **A5**: In our experiments, we trained all GPs under similar specifications, for a fair comparison of relative performance across all acquisition functions. The specifications we used are:
> - The hyperparameters $X$ are normalized to unit cube ($X \in [0,1]$), while objective and log-costs are standardized to zero-mean and unit variance.
> - The prior is set to constant mean, which defaults to a zero-mean value.
> - The covariance is set to the MaternKernel, commonly used to model smooth functions.
>
> ### **Q6: Input space**
>
> **A6**: The input space can be discrete or continuous. In our synthetic benchmarks, we optimize our hyperparameters in a continuous search space, where EEIPU makes use of memoization by using the process defined in the Memoization-based Sampling section above.
>
> Thank you again for your detailed review and observations. We hope that we were able to address your concerns, and that we successfully communicated the merit of our proposal.

---

### Author Response · Authors · 2023-11-23

We would like to apologize for the time it took us to prepare our replies. We have dedicated this time to addressing every review to clarify the novelty of our proposal by identifying the difference of setting between EEIPU and other acquisition functions, an running additional pipelines to expand on our set of experiments. We would also like to thank all reviewers for the time they put into your in-depth analysis and understanding of our paper. We dedicated the majority of this rebuttal period not only for running additional real and synthetic experiments, but also proposing a new acquisition function we refer to as MS_CArBO. We have included the adjustments in the appendix of our current draft, which we will move up to the main content of the paper, should it be accepted.

## **Problem Setup**
As helpfully suggested by a reviewer, and in order to more clearly state our contributions versus existing memoization techniques, we have included a new Problem Setup section in our appendix. If our paper is accepted, we will move this section into the main paper.

## **MS_CArBO**

MS_CArBO is a multistage version of the CArBO approach, equipped with cost awareness, cost-cooling, stagewise cost GP modeling, but does not use memoization. MS_CArBO serves as an ablation study to highlight the importance of memoization to EEIPU’s performance,  especially in pipelines where hyperparameters have little or negligible impact on stage costs.

## **New Experiments**
### **30-D Synthetic Pipeline**
To address the issue of dimensionality, we ran all functions, including MS_CArBO, through a 5-stage pipeline, optimizing 30 hyperparameters distributed across all stages. The objective GP is 30 dimensional, fitting the ensemble of hyperparameters against the objective value, as well as 5 different GPs for cost modeling.

### **Distillation Pipeline**
We added a third real experiment, which consists of a three-staged T5 transformer fine-tuning and distillation pipeline whose first stage is data preprocessing and has a single parameter, batch size. The second stage is a fine-tuning stage with three hyperparameters, namely learning rate, weight decay and number of epochs. This stage fine-tunes a pre-trained Google t5-small model with 60 million parameters. The final stage is a distillation step with four hyperparameters – learning rate, temperature, weight decay and number of epochs. This stage distills the 60 million-parameter model to a 7 million-parameter model.

## **Results**
in addition to the two new experiments, we also ran new trials on the stacking pipeline for all acquisition functions using a different seed and budget, in order to include MS_CArBO, and observe whether cost-awareness alone would have an impact on the runtime costs.
### **Number of Iterations Under a Fixed Budget**

| ACQF | Distillation | Stacking | 30-D Synthetic |
|:-------------|:--------------:|:--------------:|:--------------:|
| EEIPU | **45** | **87** | **148** |
| MS_CArBO | 44 | 63 | 104 |
| CArBO | 39 | 63 | 74 |
| EIPS | 37 | 62 | 54 |
| EI | 38| 63 | 72 |

### **Best Objective Value**

| ACQF | Distillation | Stacking | 30-D Synthetic |
|:--------|:-----------:|:--------------:|:--------------:|
| EEIPU | **0.182** | **0.732** | **-8.03** |
| MS_CArBO | 0.177 | 0.728 | -9.34 |
| CArBO | 0.181 | 0.728 | -11.79 |
| EIPS | 0.181 | 0.725 | -18.18 |
| EI | 0.173 | 0.729 | -12.77 |

These results, which are visualized in Section B of the new appendix, tell us the following:
EEIPU’s superiority is most apparent when the optimization budget is low, as shown in the stacking pipeline experiment (Appendix - Fig. 6, bottom row).
The multi-GP modeling of black-box costs delivers a more accurate estimation than a single cost GP. This is most apparent in the high-dimensional synthetic and T5 distillation pipelines, where MS_CArBO manages a more accurate modeling of stagewise costs, therefore achieving a higher number of iterations and objective value, compared to CArBO and EIPS.
In some cases, cost-awareness is able to exploit cheap areas of the search space to optimize the cost of running observations, leading to a higher number of iterations than EI, as is the case with the new synthetic and T5 distillation pipeline example.
In other cases, such as the stacking pipeline (Appendix - Fig. 6, top row), the hyperparameters have a negligible impact on the runtime costs, in which case, any cost-aware method would merely achieve similar performance to EI. EEIPU, however, implements a robust memoization approach that manages to reduce the cost of running BO iterations even in such cases, leading to significantly more iterations than any cost-modeling approach.

We hope that we managed to address all of your concerns, and that our efforts succeed in establishing the merit of EEIPU and memoization in the black-box cost setting we cover.

---

### Meta-Review · Area_Chair_2CyV · 2023-12-23

**Metareview:**

This work considers cost-aware Bayesian optimization of multi-stage pipelines in cases where the intermediate results of the pipeline (e.g., ML models / feature extractors) can be cached and re-used at a very small cost.  This problem arises frequently in real-world ML pipelines, and it is nice to see work that tackles this problem. However, this work has received rather lukewarm reviews, with the majority of reviewers agreeing that the paper implements a straightforward combination of existing ideas in the literature (fSnX, W9CP, uCj5), while not providing any new insights about the problem. Reviewer uCj5 suggests integrating ideas from the function networks paper to make use of intermediate outcomes when modeling the pipeline, as well as improvements to the acquisition function that could improve the effectiveness and scalability of the approach. Some reviewers (W9CP, gsaT) had difficulty understanding the setup and motivation for the work. Some of this was resolved in the discussion period, but perhaps including some of the stacking, segmentation, or transformer examples in the introduction could make the setup more concrete. Ultimately, reviewers were not persuaded by the author response and revisions.  I recommend that the authors use this feedback to improve upon their algorithm and narrative for any future publication.

**Justification For Why Not Higher Score:**

Too simple of an idea with no new insights.

**Justification For Why Not Lower Score:**

N/A

---

### Decision · Program_Chairs · 2024-01-16

Reject